# Immunopathological Assessment of the Oral Mucosa in Dermatitis Herpetiformis

**DOI:** 10.3390/ijerph20032524

**Published:** 2023-01-31

**Authors:** Agnieszka Mania-Końsko, Elżbieta Szponar, Aleksandra Dańczak-Pazdrowska, Monika Bowszyc-Dmochowska, Jakub Pazdrowski, Marzena Wyganowska

**Affiliations:** 1Department of Dental Surgery, Periodontology and Oral Mucosa Diseases, Poznan University of Medical Sciences, 70, Bukowska St., 60-812 Poznań, Poland; 2Department of Dermatology, Poznan University of Medical Sciences, 49, Przybyszewskiego St., 61-701 Poznań, Poland; 3The Maria Skłodowska-Curie Greater Poland Cancer Center, Department of Head and Neck Surgery, Poznan University of Medical Sciences, Garbary St., 61-701 Poznań, Poland

**Keywords:** dermatitis herpetiformis, oral mucosa lesions, immunodiagnostics, direct immunofluorescence

## Abstract

Dermatitis herpetiformis (Duhring’s disease, DH) is a chronic blistering cutaneous condition with pruritic polymorphic lesions, consisting of vesicles, papules or nodules and erythema, found predominantly on the extensor surfaces of the limbs, buttocks, and neck. Diagnosis is based on characteristic clinical and immunopathological findings. Oral manifestations of DH have rarely been described. The aim of the study was to evaluate IgA, IgG, IgM and C3 complement deposits in the oral mucosa in DH patients. Direct immunofluorescence (DIF) was performed on the oral mucosa specimens collected from 10 DH patients. Biopsy was taken in a local anesthesia from perilesional site from the buccal mucosa and then preserved in a standard procedure using polyclonal rabbit IgG, IgA, IgM and C3 antibodies. Granular IgA and C3 deposits were found in 6 patients (60%), and in 3 subjects (30%) the result was indeterminate. Significant fluorescence of the deposits along the basement membrane was observed in 2 patients, moderate fluorescence in 3 patients, and in 4 cases the result was indeterminate. C3 deposits were found in 5 subjects (50%), 3 of them being moderate and 2 indeterminate. No IgM and IgG deposits were detected in the collected buccal mucosa specimens.

## 1. Introduction

Dermatitis herpetiformis (Duhring’s disease, DH) is a chronic autoimmune, blistering cutaneous condition characterized by polymorphic lesions such as vesicles, erosions, erythematous patches, urticaria and nodules [1,2,3,4,5,6], that are usually found symmetrically on the extensor surfaces of the limbs, buttocks, and neck [2,4,5,6,7,8,9,10,11,12]. These lesions are accompanied by intense itching of the skin [2,4,10,11,12,13,14]. DH occurs with an asymptomatic sensitivity to gluten that is contained in wheat, oats, barley, and rye grains [1,3,6,14,15,16]. 

When applying gluten-free diet, dermatitis herpetiformis skin lesions heal. They reappear with gluten introduction [2,6,11,12,13,15].

The incidence of DH in Europe ranges between 0.4 to 3.5 per 100,000 people [6]. It is rarely observed in African and Asian countries, which may be due to dietary habits in these regions, i.e., high intake of rice, tapioca, amaranth and low intake of wheat and rye [1,11].

Clinical signs on the oral mucosa in DH patients have rarely been observed. It included erythematous patches, petechiae, nodules or erosions located mostly on the tongue, buccal, labial, and alveolar mucosa, and the soft palate [11,17,18,19]. 

Dermatitis herpetiformis can be confirmed based on 2 out of 3 clinical signs which include typical skin lesions, the presence of granular IgA deposits in the papillae peaks close to the dermal-epidermal border, and the presence of the IgAEMA serum and/or TGA antibody [11,12,20].

The diagnosis is based on immunopathological findings. Skin bioptats are evaluated with a direct immunofluorescence test (DIF) and the presence/absence of granular IgA deposits in the papillary dermis and along the dermal-epidermal junction. Deposition of IgG, IgM and C3 may also occur [1,2,4,6,10,11,12,13,21,22,23,24,25]. Histopathological examination of a small bowel specimen shows varying degrees of villous atrophy and lymphocytic infiltration [2,6,13,16,25].

Few studies evaluating specific antibodies in the oral mucosa specimens in DH individuals have been published to date.

Given a high prevalence of skin lesions, a wide variety of clinical manifestations, and a small number of oral mucosal examinations in patients diagnosed with DH, it seemed interesting to conduct a study to determine the presence of immunoglobulin deposits in the oral mucosa of patients with dermatitis herpetiformis.

The premise of the study was to control if the same immunoglobulin deposits are present in the oral mucosa and in the skin, and if examination of an oral mucosal slice could contribute to the diagnosis of dermatitis herpetiformis, where lesions occur on the oral mucosa, only, preceding the skin symptoms.

This study aims at immunopathological assessment of the oral mucosa bioptats in DH patients whose diagnosis was confirmed by the DIF result from the dermal specimens.

## 2. Materials and Methods

The study group consisted of 10 subjects (6 female and 4 male) with DH diagnosed with a DIF test of the dermis. The specimens were taken from the clinically intact buccal mucosa in a local anesthesia (2% lidocaine) with forceps and no need of applying stitches. Clinical oral mucosa examination and biopsies were taken in the Department of Oral Mucosa Diseases, University of Medical Sciences in Poznan. Collected tissue samples were then transported, within few hours, in plastic containers with a saline solution to the Laboratory of Skin Immunopathology and Histopathology of the Department of Dermatology, University of Medical Sciences in Poznan. The samples were refrigerated and cut in a cryostat. Polyclonal FITC-conjugated rabbit anti-human IgG, IgM, IgA and C3 antibodies were used as reagents (Dako, 1:100 dilution in a phosphate buffer—PBF, pH = 7.6) and the specimens were then incubated with the reagents in a humid chamber and in a room temperature for an hour. The slides were subsequently washed in the PBF, a drop of 10% PBF glycerin solution was added, and the glass covers were placed over the specimens. Fluorescent microscope (Zeiss) and a digital camera (Olympus) were used for the immunopathological assessment of the specimen slides. The images obtained by the digital camera were not photo edited.

The results were presented in a semi-quantitative scale, describing the fluorescence intensity from “−” to “++”. A negative result was the reference value.

The study design was approved by the local Ethics Committee (Resolution No. 355/10). Written informed consent was obtained from all the study participants.

## 3. Results

Granular IgA and C3 deposits were found in 6 patients (60%), and in 3 subjects (30%) the result was indeterminate. Significant fluorescence of the deposits along the basement membrane was observed in 2 patients, moderate fluorescence in 3 patients, and in 4 cases the result was indeterminate. One patient had no IgA, IgM, IgG and C3 deposits detected.

C3 deposits were found in 5 subjects (50%), 3 of them being of moderate fluorescence and 2 were indeterminate. No IgM and IgG deposits were detected in the collected buccal mucosa specimens. The results of DIF tests in the study participants are shown in Table 1.

Selected microscope images of the oral mucosa bioptats are shown in the Figure 1, Figure 2, Figure 3 and Figure 4.

## 4. Discussion

In recent years, particular attention has been paid to the association of skin and gastrointestinal diseases with oral health [11]. Due to a small number of studies on the immunological evaluation of the oral mucosa in patients with dermatitis herpetiformis, studies assessing the presence of immunoglobulin deposits in the oral mucosa with direct immunofluorescence (DIF) have been undertaken.

We assessed the presence of immunoglobulin deposits of IgA and C3 along the basement membrane of the oral mucosa epithelium the test can be helpful and used as a complementary test in the diagnosis of DH in patients with oral mucosal lesion and to exclude other oral mucosa diseases.

To the best of the authors, knowledge, only few studies have been published to date on the immunological assessment of the oral mucosa in DH (4 papers) [17,19,26,27].

In our study, we analyzed 10 buccal mucosal sections from 10 patients with dermatitis herpetiformis. Six patients (60%) showed granular IgA deposits along the basement membrane of the oral mucosal epithelium. C3 complement component deposits were detected in 5 patients (50%). Simultaneous IgA and C3 deposits were found in 5 patients. No IgG and IgM deposits were detected. The group we studied was balanced in terms of age and gender.

These study results are similar to those obtained by the authors. A comparison of those studies is shown in Table 2.

Nisengard et al. performed biopsies from the buccal mucosa and gingiva in 14 DH patients. DIF showed IgA deposits in 6 out of 13 patients (46%) and IgG deposits in 2 out of 13 patients (15%) in the bioptats from the buccal mucosa. In the tissue collected from gingival IgA, deposits were found in 3 patients (21%) [19]. Harrison et al. examined 7 subjects with DH and coeliac disease. They observed granular IgA and C3 deposition in all the patients, but no IgG and IgM deposits were detected in the bioptats taken from the buccal mucosa [26]. The study carried out by Hietanen et al. also included biopsies from the buccal mucosa and DIF test of the specimens taken. It disclosed IgA deposits in all the 7 patients with DH, and in 3 of them C3 deposits were found [27]. Fraser et al. collected bioptats from the oral mucosa of 4 DH patients. In 2 subjects (50%) granular IgA deposits were detected and in 1 patient IgG deposits were found in a DIF test, while other sections showed no evidence of IgA and IgG deposits [17].

The results of our study are comparable in terms of the number of patients and the presence of immunoglobulin deposits in the oral mucosa with the few studies by other authors to date. 

The studies to which we referred were carried out decades ago, and so far, no similar studies have been performed where immunoglobulin deposits in the oral mucosa were assessed in patients with DH, which offers great prospects for future studies on a larger group of patients.

## 5. Conclusions


Granular IgA and C3 deposits were found along the basement membrane of buccal mucosa epithelium in most of the subjects.IgA and C3 deposits detected with DIF in the oral mucosa specimens may be a supplementary assay in DH diagnosis. IgG and IgM deposits were not present in buccal mucosa bioptats in DH subjects.


The clinical diagnosis of dermatitis herpetiformis is based on a complete history, a detailed examination of the oral cavity and skin, while a proper diagnosis confirming DH requires biopsy and immunological testing. We found IgA and C3 deposits detected by direct immunofluorescence in apparently unaltered oral mucosa confirmed DH and may be a complementary study to the diagnosis of DH, where oral lesions often precede skin symptoms.

As the patient group was small, it would be appropriate to consider this study as a pilot study.

## Figures and Tables

**Figure 1 ijerph-20-02524-f001:**
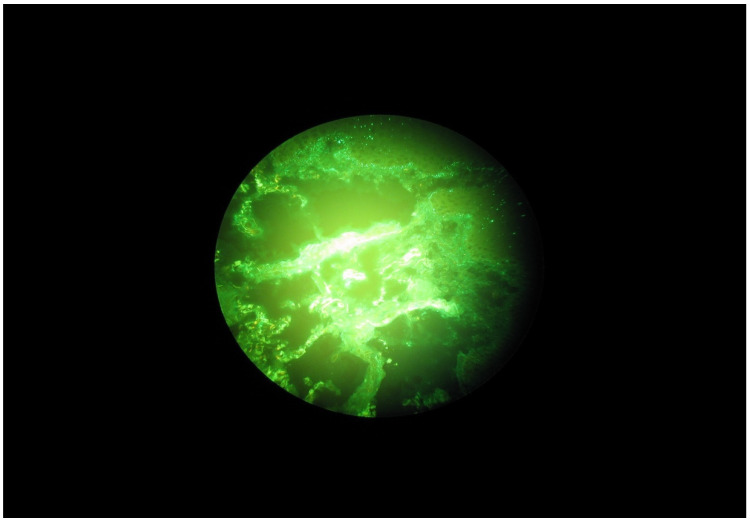
Granular C3 deposits along the basal membrane of the oral mucosa in a DH patient.

**Figure 2 ijerph-20-02524-f002:**
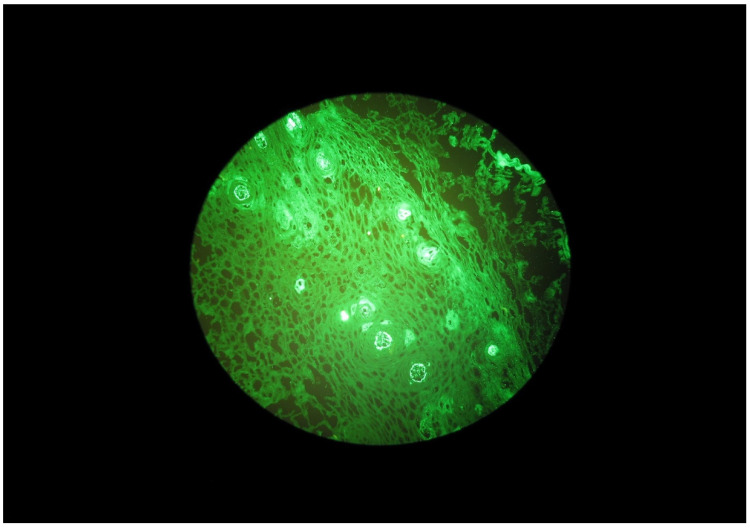
Granular IgA deposits beneath the basal membrane of the oral mucosa in a DH patient.

**Figure 3 ijerph-20-02524-f003:**
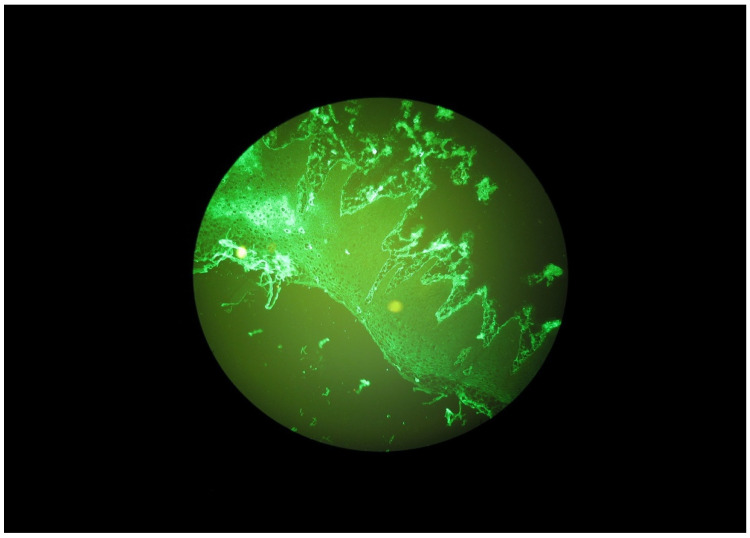
Granular IgA deposits along the basal membrane of the oral mucosa in a DH patient.

**Figure 4 ijerph-20-02524-f004:**
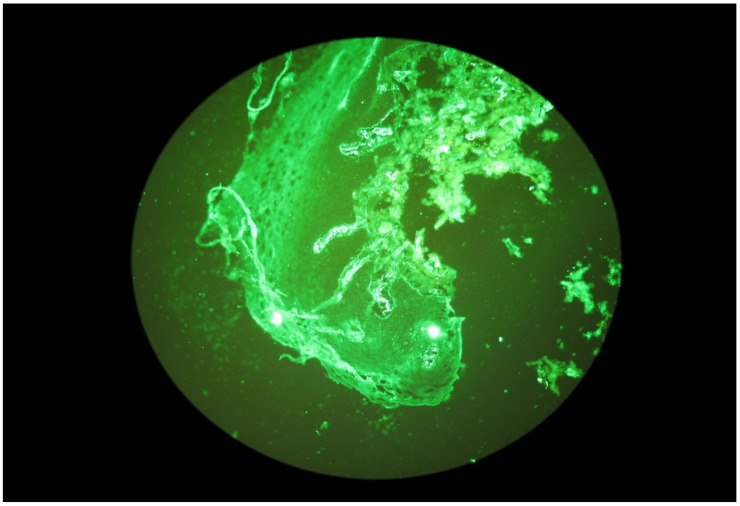
Granular IgA deposits dispersed in stromal cells.

**Table 1 ijerph-20-02524-t001:** DIF results of the oral mucosa bioptats in the study participants with DH. DIF—direct immunofluorescence, f—female, m—male, +/− indeterminate DIF result, + moderate fluorescence intensity, ++ significant fluorescence intensity, − no deposits detected, Ig—immunoglobulin.

No.	Sex (m/f)	IgA	IgM	IgG	C3
1	f	−	−	−	−
2	m	+/−	−	−	+/−
3	m	+/−	−	−	−
4	f	+	−	−	−
5	m	+/−	−	−	+
6	f	+/−	−	−	−
7	m	++	−	−	+/−
8	f	+	−	−	+
9	f	+	−	−	−
10	f	++	−	−	+

**Table 2 ijerph-20-02524-t002:** Immunoglobulin and C3 deposits in the oral mucosa according to the present study and other authors’ studies.

No	Author	Number of Study Participants	IgA Presence	C3 Presence
1	Russell et al.	13	6 (46%)-buccal mucosa 3 (21%)-gingiva *	none
2	Harrison et al.	7	7 (100%)	7 (100%)
3	Hietanen et al.	7	7 (100%)	3 (43%)
4	Fraser et al.	4	2 (50%)	none
5	Mania-Końsko et al.	10	6 (60%)	2 (20%)

* in a group of 14 patients.

## Data Availability

The date is available on demand.

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
