# Peer review of "Immunopathological Assessment of the Oral Mucosa in Dermatitis Herpetiformis"

_ijerph, 2023, doi:10.3390/ijerph20032524_

Round 1
Reviewer 1 Report
As the group of patients is small, I suggest that you add "pilot study" in the title. The results are compared with studies also carried out on a small group of patients, so the comparison is fair. I would suggest complete the conclusions with an explanation of the clinical importance of the immunopathological investigations.
Author Response
Dear Editor,
We appreciate the time and efforts by the editor and referee in reviewing this manuscript. We have addressed all issues indicated in the review report. We hope that the revised version will satisfy the reviewers and meet the journal publication requirements.
Response to the Reviewer’s #1 comments
Comment 1 : As the group of patients is small, I suggest that you add "pilot study" in the title. The results are compared with studies also carried out on a small group of patients, so the comparison is fair. I would suggest complete the conclusions with an explanation of the clinical importance of the immunopathological investigations.
Response: Thank you for this comment. As suggested, ''pilot study'' has been added to the conclusions.
Comment 2: I would suggest completing the conclusions with an explanation of the clinical importance of the immunopathological investigations.
Response: Thank you for this suggestion. Conclusions were supplemented by an explanation of the clinical significance of immunopathological tests.
(The clinical diagnosis of dermatitis herpetiformis is based on a complete history, a detailed examination of the oral cavity and skin, while a proper diagnosis confirming DH requires biopsy and immunological testing. We found IgA and C3 deposits detected by direct immunofluorescence in apparently unaltered oral mucosa confirmed DH and may be a complementary study to the diagnosis of DH, where oral lesions often precede skin symptoms.)
Reviewer 2 Report
-The present paper faces a topic already extensively addressed in the literature. The research protocol used is badly defined. Therefore, major revisions are needed.
-English revision is required by a native English speaker.
-The "Introduction" section is insufficient. It is necessary to report the state of the art on the topic addressed, and what the literature says about the chosen topic. A detailed analysis of the gaps present in the literature is missing. Please, report the outcomes extracted from the literature in reference to the objectives of your study.
-It is necessary to insert at least one null hypothesis at the end of the introduction section.
-What about the criteria of inclusion of the patient included in the present study?
-The "Discussion" section should begin by stating whether the null hypothesis has been accepted or rejected.
-The "Discussion" section itself is unacceptable as it stands now. There are no sufficient explanations to justify the results obtained. Furthermore, comparisons with the results present in the literature are missing and a paragraph regarding future study perspectives is missing.
-A detailed paragraph regarding the limitations of this clinical study is mandatory in the "Discussion" section.
-The "Conclusions" section cannot be so schematic and short. It needs to be enriched and discursive.
-The list of References is scarce and many recent papers are missing. Please update the reference list.
-Pay more attention in writing the references shown in the Reference list. Indeed, little attention seems to have been paid to the authors' guidelines, especially in reporting the name of the journal related to each reference (e.g., "British Journal of Dermatology" or "European Journal of Oral Sciences" need abbreviation).
Author Response
Dear Editor,
We appreciate the time and efforts by the editor and referee in reviewing this manuscript. We have addressed all issues indicated in the review report. We hope that the revised version will satisfy the reviewers and meet the journal publication requirements.
Response to the Reviewer’s #2 comments
Comment 1: The present paper faces a topic already extensively addressed in the literature. The research protocol used is badly defined. Therefore, major revisions are needed.
Thank you for the comment. The topic of dermatitis herpetiformis and the study of immunoglobulin deposits located in the papillae peaks, close to the dermal-epidermal border, is widely reported in the literature. Studies of the presence of immunoglobulins in the oral mucosa have only been presented by single authors. To date, only four authors have attempted to assess the presence of immunoglobulin deposits in the oral mucosa. Oral mucosal lesions are often the first symptoms, preceding skin lesions, and therefore these studies may be helpful in the earlier diagnosis of dermatitis herpetiformis.
Comment 2: -English revision is required by a native English speaker.
Thank you for the comment. The work has been checked and corrected in terms of language
Comment 3: -The "Introduction" section is insufficient. It is necessary to report the state of the art on the topic addressed, and what the literature says about the chosen topic. A detailed analysis of the gaps present in the literature is missing. Please, report the outcomes extracted from the literature in reference to the objectives of your study.
Thank you for the comment. As previously mentioned, there is a lack of current research on the presence of immunoglobulin deposits in the oral mucosa.
As suggested by the reviewer, the current state of knowledge regarding dermatitis herpetiformis was corrected in the introduction.
The aim of the study was to assess the presence of immunoglobulin deposits in the oral mucosa.
Comment 4: It is necessary to insert at least one null hypothesis at the end of the introduction section.
Thank you for the comment. Examination of a skin section by direct immunofluorescence (DIF) from non-transformed areas reveals the presence of granular IgA deposits in the papillary peaks and along the dermal-epidermal junction. The deposits may be accompanied by C3 complement components, IgG and IgM deposits. The premise of the study was to see if the same immunoglobulin deposits are present in the oral mucosa as in the skin, and whether such a study could be a complementary test in the diagnosis of dermatitis herpetiformis when oral mucosal lesions are present.
Comment 5: What about the criteria of inclusion of the patient included in the present study?
Thank you for the comment. In the objectives of the study, it was noted that the criterion for examining oral mucosal sections with immunological evaluation was to study only patients with a diagnosis of DH, confirmed by skin DIF examination.
Comment 6: The "Discussion" section should begin by stating whether the null hypothesis has been accepted or rejected.
Thank you for the comment. As proposed, confirmation of the null hypothesis was placed at the beginning of the discussion.
Comment 7: The "Discussion" section itself is unacceptable as it stands now. There are no sufficient explanations to justify the results obtained. Furthermore, comparisons with the results present in the literature are missing and a paragraph regarding future study perspectives is missing.
Thank you for this valuable comment. The discussion is extended and the results of the study are explained.
The study we referred to was conducted several decades ago AND no similar study has been performed so far where immunoglobulin deposits in the oral mucosa were assessed in patients with DH, which offers great prospects for future studies on a larger group of patients.
The group we studied was balanced in terms of age and gender.
Comment 8: A detailed paragraph regarding the limitations of this clinical study is mandatory in the "Discussion" section.
Comment 9: The "Conclusions" section cannot be so schematic and short. It needs to be enriched and discursive.
Thank you for the comment. Conclusions were developed and a suggestion from the first reviewer , to consider the work as a pilot study due to the small study group, was added.
Comment 10: The list of References is scarce and many recent papers are missing. Please update the reference list.
Thank you for the comment. The list of references has been updated
Comment 11: Pay more attention in writing the references shown in the Reference list. Indeed, little attention seems to have been paid to the authors' guidelines, especially in reporting the name of the journal related to each reference (e.g., "British Journal of Dermatology" or "European Journal of Oral Sciences" need abbreviation).
Thank you for the comment. After reviewing the literature, the notes dedicated to the authors' guidelines have been corrected. Where journal abbreviations are required, these have been corrected.
Round 2
Reviewer 2 Report
I am Happy with the changes made by the Authors